# Inclusive Economic Sustainability: SDGs and Global Inequality

**Arno J. van Niekerk** 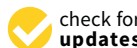

Department of Economics and Finance, Faculty of Economics and Management Sciences, University of the Free State, Bloemfontein 9301, South Africa; niekerka@ufs.ac.za; Tel.: +27-51-401-3271

**Abstract:** In view of the 2020 global health crisis and its repercussions on the global economy, the need to redirect conventional economic thinking towards securing global economic sustainability is most critical. The Sustainable Development Goals (SDGs) are a significant move in this direction. However, in the past few years, a clearer understanding of inclusive economics and sustainability indicators have progressed our ability to reduce economic exclusion, chiefly represented by global inequality. Collective wellbeing within the "global village" is shaped largely by these avenues/directions, thus presenting the question: can an improved combination of sustainability priorities be identified that would substantially enhance countries' adoption of the SDGs? New, inclusive paths to economic progress are essential to a world economy in crisis recovery mode. The aim of the paper is to qualitatively identify key indicators from these different directions to, collectively, address some of the most significant drivers of global inequality, thus improving the adoption rate of the SDGs. As its main contribution, the study found that for economic inclusivity to realistically reduce global inequality its full integration into three areas is necessary: business models, public policy and community development. This should also be supported by "social covenants" to facilitate improved SDG adoption by countries.

**Keywords:** inclusive economics; sustainability; SDGs; inequality; wellbeing; genuine progress

---

## 1. Introduction

In a time when the world is arguably facing one of its most severe challenges due to the coronavirus-related lockdown restrictions, it is the ideal moment to rethink and address the systemic failures of our current economic framework. While this severe health crisis is just another crisis among many other financial crises with global ramifications over the past 30 years, which have exposed the flaws of neoliberal capitalism, it calls for a drastic ideological redirection [1]. Due to its higher degree of economic interdependence than perhaps ever before, the global community's shared socioeconomic difficulties present an opportunity to now shape and fully implement inclusive economic principles on the foundation laid by the already shared Sustainable Development Goals (SDGs).

During, particularly, the past 20 years, which first involved the Millennium Development Goals (MDGs) of 2000–2015 and now the SDGs (2015–2030), much progress has been made in cultivating new economic thinking [2]. The eight goals of the MDGs were expanded to 17 goals under the SDGs. Economic sustainability and reducing economic inequality has been at the forefront of the agenda. In this context, institutional economic theory aids in explaining that market deficiencies germinate from diverse interactions between individuals, firms, states and social norms. Furthermore, neorealist economic theory brings into perspective here the distorting effects of bounded rationality (economic choices limited by information availability, time, etc.) and dominant role players (especially usurpers of power). Since the matter of "exclusion" (of both people and nature) is arguably the root issue in dealing with sustainability, the real vacuum has been the absence of a proper "inclusive"

economic framework. Limited literature—both theoretical and empirical—exists on this topic, of which the majority is narrowly focused on inclusive growth. This leaves a gap in the literature in terms of the broader understanding of inclusive economics. The latter's focus is on how to include more people in productive processes (countering inequality) and how to include better management of natural resources in the economic cycle to ensure not only sustainability but also genuine economic progress. Gradually an inclusive economic framework evolved (and is evolving) that can add significant value to the implementation of the SDGs and their country-specific effectiveness (i.e., their ability to ensure real economic development) [3]. This is an important topic given the growing need for the SDGs to be more successfully implemented than the MDGs were. A key question is, thus: what would a more integrated framework of combined sustainability priorities consist of, which would increase their effectiveness and result in increased adoption of the SDGs by countries? In answering this question, the study becomes a qualitative assessment—following a theoretical literature review approach—of a possible symbiosis between inclusive economic criteria, the SDGs and sustainability indicators towards addressing global inequality.

## 2. Conceptual Framework

As history would possibly show, the wellbeing threat of the 2020 "global lockdown" due to COVID-19 brought a fundamental reorientation of economic realities, concepts and priorities. After the 2007–2009 global financial crisis, this is the first (and probably worst) fully global test to the strength of the "global village". It has reinforced the sense of shared and collaborative responses to common existential threats and the prosperity of our shared future. To our advantage, inclusive economics (IE) or economic inclusivity has emerged as a progressive disciplinary perspective on how to achieve inclusive development, inclusive growth, inclusive governance and a circular, more sustainable, economy [3]. IE represents the empirical merging of the interdependent reality of being human and the increasingly interdependent nature of the global economy, with collaborative networking (supported by technology) as the common denominator.

### 2.1. Inclusive Economics

As a concept that is still evolving, IE can be described as a holistic economic framework focused on optimizing collective wellbeing through preserving human capital, ecological capital and shared social norms in a balanced and innovative way so as to create sustainable and equitable opportunities, access and benefit sharing. In this integrated framework, the aim is to (re)shape a circular economy of care through enabling technologies, inclusive business, civic participation and inclusive economic policies—all geared towards finding ways to increase the participation of economic role players. In effect, it is a synthesis between collective wellbeing (not just welfare) on the one hand and classical capitalism (profit-driven rational choice theory and Pareto optimality (growth)) on the other hand, which creates value for all stakeholders as sustainability becomes the driver of innovation. It brings a paradigm shift in economics from self–interest as the primary motivation to shared interest. This shift stimulates investment in human wellbeing (optimizing community social capital), promotes even non-income aspects of wellbeing and places value on the net impact of production, leading to a "shared commitment" (or social covenant) taking shape where inclusion bring greater equality (and preservation) of conditions.

IE essentially involves a balanced "inclusion" of mainly three aspects in the economic framework: people, nature and ethics. In addressing the "exclusion" flaws of neoliberal capitalism, IE reinforces a new emphasis in economics by giving the highest priority to (1) reducing poverty and unemployment (i.e., people's exclusion from the economy); (2) the environmental responsibility of economic processes (i.e., reduce wasting of natural resources); and (3) moral obligation and value-driven profit making (i.e., preventing the exclusion of ethics in economics). Importantly, IE is oriented towards addressing two fundamental changes that have taken place in economies worldwide. First is the changing structure of economies as new sectors emerge (e.g., highly specialized private medical and financial services) and

traditional sectors such as manufacturing are radically altered [4]. Even the relationships between the paid and unpaid economies and between the formal and informal sectors are changing. Second is a reintroduction of ethics and morality, and culture and context, into economics, in recognition of changing social values assigned to economic performance [5]. Social justice, creating real opportunities for the poor and ensuring environmental sustainability are of growing public interest as the morals and ethics of growth are under the microscope, along with a more pragmatic, reality-based approach to economics. Consequently, in this context, the aim of genuine economic progress is central to IE.

One measurement for this is the genuine progress indicator (GPI) that has been developed as an alternative—or at least a complementary—measure to gross domestic product (GDP). The GPI incorporates key elements of human wellbeing, including a wider range of factors than GDP. A combination of 25 economic, social and environmental primary research indicators are captured (as plusses and minuses) to determine the sustainability of income growth [6]. Overall progress in these areas will, according to the GPI, determine whether development is sustainable and inclusive. Some of the variables include income inequality; net capital investment; pollution abatement; climate change; crime; underemployment, etc. The following mode of calculation is also used to arrive at GPI:

$$GPI = Cadj + Gnd - W - D - E - N$$

With each component consisting of sub-variables, the six main variables are $Cadj$, personal consumption expenditure adjusted for income inequality; $Gnd$, all non-defensive government expenditures (e.g., building schools); $W$, non-monetized contributions to welfare (e.g., volunteer work); $D$, defensive private expenditure (e.g., insurance); $E$, the costs of environmental degradation (e.g., deforestation); and $N$, the depreciation of the natural capital base (e.g., non-renewable consumption). Comparing this with GDP, various studies in the United States, New Zealand, Australia, China, Singapore, France and others have been done to determine countries' GPI [7–9]. What is clear from almost all of them is that the gap between GDP growth and GPI decline is increasing, albeit at varied degrees. This makes growth unsustainable, and, especially in light of the 2020 global health crisis, a rather impossible reality if the human side of the economy is not sufficiently accounted for and attended to. This brings GPI strongly into the picture to ensure inclusive economic priorities are elevated, such as poor people's income rising faster than that of the non-poor; growth to be non-discriminatory and disadvantage-reducing; more equitable access to goods and services, not only determined by ownership (e.g., open-source, Uber, Airbnb, etc.); protecting and using natural resources sparingly; GPI per capita to rise in tandem with (or faster than) GDP per capita; and non-market contributions to the economy to be valued better.

While the GPI—just like GDP—is not a perfect measure of the performance of economies or a complete measure of welfare, and is still being refined, it represents a meaningful departure (along with other wellbeing measures) from the growth-only mantra. As pointed out by Berik [10], the GPI "was the first aggregate welfare indicator to incorporate an inequality adjustment that deducts the cost of inequality from personal consumption expenditures" (p. 79). Its inclusive nature is reflected by it being both a measure of economic performance and of social progress. In sum, all this contributes positively to a move towards what The Aspen Institute [11] calls "inclusive capitalism"—an economic system in which the benefits of growth are broadly shared, creating more opportunities for all people to improve their economic status.

## 2.2. Economic Sustainability

A second key concept in this paper is economic sustainability. The latter proceeds from traditional "environmentalism" by recognizing the importance of providing secure, long-term employment without endangering the health of ecosystems [12]. Sustainability, in economic terms, is understood as the economy's ability to maintain itself and continue to operate without jeopardizing over time the very purpose of its existence: managing resources from nature and for people. It refers to practices

that sustain long-term economic growth without adversely impacting environmental, social and cultural aspects of community. Since markets have historically failed to protect the environment, the balancing of economic interests and ecological interests remains a challenge. This challenge is exacerbated by, particularly, the "internal conflict" within economic interests, namely between profit and equity (social justice), resulting in the following pillars being identified for economic sustainability: economy (employment), environment and equity [13]. These three have competing goals and interests born out of concern for each of them. As such, the concept of "sustainable development" has emerged with the objective of conflict resolution between these pillars, working with strategies to bring the level of sustainability closer to the (ideal) sustainable human–environment system [14]. In this regard, sustainability seeks a context in which the legitimate interest of all parties can be satisfied—to a greater or lesser extent. To assist with this, Edwards [12] added a fourth "E", education, to the three pillars, emphasizing that education is the catalyst for helping people understand the dynamic nature of the interrelationship of the three Es. For example, the fair, careful and equitable distribution of resources is not just essential for community wellbeing, but ethical, and respectful of ecological limits.

Possibly, the landmark document that set the tone for sustainable development was the 1987 Brundtland Report [15], which famously described it as "development which meets the needs of the present without compromising the ability of future generations to meet their own needs" (p. 16). The term "development" was often taken to refer to the processes of economic and social change in, predominantly, the developing countries. This report moved the focus towards the integration of environment, development and social justice in both rich and poor countries. It built on a more inclusive "new science" that was emerging from the mid-twentieth century, which Herbert Marcuse [16] described as "our desire to live in harmony rather than in conflict with nature" (p. 32).

Hence, instead of sustainability just being seen as an "add-on", it came to be viewed as a necessity in a more integrated context in the economy. In light of this, Costanza et al. [17] define sustainable development as "development that improves the quality of human life while living within the carrying capacity of supporting ecosystems" (p. 12). The United Nations [18] regards it as the harmonization of economic growth, social inclusion and environmental protection in order to raise living standards. Over time it became clear, however, that significant difficulties and trade-offs complicated this harmonization.

This led to the concept of "inclusive development" emerging strongly after the global financial crisis. It started placing greater emphasis on the poorest and most marginalized [19]. Inclusive development went further by including in its focus interactions between different levels: individual, governmental and international relations. Progress within the human rights domain—especially regarding basic needs (e.g., water and sanitation services)—further sharpened its focus on consciousness and deeper understanding of inclusive human development. According to Gupta et al. [20], inclusive development "calls for direct democracy (the exercise of civil, civic and political rights) and the distribution of amenities (e.g., health, education and infrastructure) with a view to enabling participation by all in these amenities" (p. 36). Reproducible stocks in society, such as natural and human capital, institutions, population and time then become the primary focus instead of only flows (income). This recognizes that positive economic growth is compatible with a negative value for inclusive wealth and that it is primarily human capital that counterbalances losses in natural capital. It promotes investment in renewable capital to reproduce wellbeing in society. Keeping the wellbeing of future generations in mind, as well as the management of much-needed ecosystems, the focus is less on the growth rate and more on investing in human wellbeing. Inclusive development is an adaptive learning process that responds to change and new risks of exclusion and marginalization. Signifying its scope, Gupta et al. [21] identified six reasons for inclusive development:

- Normative: Reducing indignities and empowering the poor, justified by moral values;
- Legal: Institutionalizing social norms and advancing minimum acceptable conditions for all human beings grounded in their human rights;

- Economic: Raising participation in production and consumption processes to enhance people's material, relational and human wellbeing, including that of future generations;
- Security: Reducing social conflict over resources (and promoting innovative distribution) and enabling the poor to have access to legal means of survival and live in safety;
- Political/democratic: Taking account of the needs of the poorest as part of the electorate and engaging all in decision making (procedural justice) and in sharing resources and prosperity (distributive justice);
- Relational: Limiting poverty resulting from the actions of others and/or the exploitation of the resource base by strong economic role players (e.g., monopolies).

*2.3. Global Inequality*

A third key concept in the context of this paper is global inequality. We live in a world where there exists a paradoxical "development gap" between unprecedented opulence and staggering deprivation [22]. This, which Stiglitz [23] calls "scarcity in an age of plenty", has seen the standard of living of over half a billion people, in countries such as China, India and Brazil, raised through a vibrant world economy over the past three decades, yet global inequalities increased exponentially [24]. How is it possible, therefore, that in a technologically advanced 21st century, where we have solved incredibly complex problems, we have not found answers to the persistence of extreme poverty? This makes the study of global inequality as a multidimensional concept and reality so important. At issue here is, particularly, how it excludes people from mainstream economic activity, especially the formal sector. Bourguinon [24] defines global inequality as "the level of inequality between all inhabitants of the world" (p. 9). Defined in relative terms, it is seen as inequality between the standards of living of individuals in the global population as a whole. In absolute terms, it is seen as the extent of global poverty itself (the percentage of the world population living below the poverty line of 1.25 dollars per day). The complexity of global inequality arises from its multiple facets.

Firstly, it consists of a complex combination of inequalities across nations and inequalities within nations. The former considers the per-capita income gap between the richest and poorest countries, which has progressively increased over the past 200 years from a ratio of 3:1 to almost 50:1 [25]. While global per capita income trebled between 1960 and 2000, over 100 countries have experienced a decline in per-capita income since the 1980s. The latter, using the Gini Index, shows a world where, in 2005 for instance, 53 countries (containing 80% of the global population) experienced rising inequality, compared to only nine countries (containing 4% of the world's population) where a narrowing of inequality occurred [26]. Even developed countries like the United States have experienced a sharp rise in inequality, registering a Gini coefficient of 0.41 in 2016 compared to 0.38 in 1991, with the richest 10% earning $70,000 on average per year in 2015 and the poorest 10% earning $4500 per year (Current Population Survey) [15]. In global terms, the richest 600 million people (living standard above $25,000) were 92 times richer than the poorest 600 million people (with an average disposable income of $270 per year) in 2015 [24]. Reporting on global inequality, the UNDP [27] points out that the growth of global income distribution between 1980 and 2016 to the top 1% economic elite in rich and poor countries was extremely high: over 200%. During this period, the top 1% received 27% of income growth while the bottom 50% received a mere 12%. This gives credence to the alarming claim by Oxfam [28] that, in 2018, the world's 26 richest people owned as much as the poorest 50% of the world's people (3.8 billion people).

No matter how you measure it, the numbers show a world that is extraordinarily unequal compared to any national norm. Global inequality is above the highest levels of inequality seen in highly unequal countries (global Gini coefficient was 0.7 in 2008) and arguably above what a national community could bear without risking a major crisis. Of serious concern, in light of globalization, are the increasing spillover effects between inequality between countries and that within countries [29]. The negative effects of excessive inequality on economic efficiency and individual welfare are now expected to

be exacerbated by the global coronavirus crisis. As such, the continued exclusive appropriation of economic progress by a small elite will create excessive risk to the stability of societies.

The second reason for global inequality's complexity is the fact that there are other dimensions to inequality and poverty than income. Pointing to the interconnected nature of global inequality, Greig et al. [25] highlight a number of social and political inequalities: differences in access to health services, basic infrastructure and the legal system; different educational levels; and people's ability to participate in public decision making, to name a few. Important, here, is the debate between the individualist approach to inequality and the structuralist approach. The former follows a neoliberal (or rational choice) approach and focuses on the endowment factors within poorer countries (i.e., what they lack in terms of development) and individual human agency (i.e., people determine their economic situation by their choices). Each country is assessed as a self-contained unit with the focus on aligning countries' development policies and outcomes with the "form of development" and the growth trajectory of now-developed countries. In this endogenous approach, a distinction is made between the "deserving" and "undeserving" poor [30]. The former refers to those who cannot participate in economic activity due to uncontrollable factors, such as disability, accident, infirmity, and old age. The latter include voluntary indigent people choosing to be poor due to their immorality or character defects: drunkenness; lassitude; work shirking; promiscuousness, crime; etc. Not to be excluded here is the role of what Birdsall [31] calls "constructive inequality", whereby it may create positive incentives at the micro level, such as stimulation of individual effort, productivity and innovation as better moral choices are made for change.

Of concern, however, is "destructive inequality", which entrenches social exclusion. In line with this, the structuralist perspective is more "exogenous" and focused on identifying and transforming the very structures that connect deprivation to opulence. For instance, powerful actors are of concern in so far as their dominance undermines the development of subordinate people (within countries) and nations [25]. Examples include inequality of opportunities and security; maldistribution of resources; limited social mobility; preventing access to land; gender inequalities; discrimination against race, class, age, etc.; imposing preferences (e.g., low wages on workers); and coercively limiting people's economic choices. Essentially, these consist of any influences flowing from power differentials in social relations that impede social equality.

Of further concern is how the exploitation of unequal power relations aggravates the already-unequal playing field characterized by the digital divide; vicious poverty cycles; restricted access to water or credit; limited human capital; natural resource depletion; information asymmetry; segmented labor markets; social polarization; non-delivery of essential services, etc. As stated by Greig et al. [25], "inequalities are not simply carefully constructed measurement scales but complex webs of dynamic social relations that privilege some while constraining the life chances of others" (p. 27). Since the goal to be achieved is not "economic equality", as it is impossible, the joint goals are equality of opportunity (social justice) and equitable economic outcomes. The actual issue here is how wealth is generated and, more specifically, the nature of economic growth in terms of the extent to which it is "at the cost" of the poor and "at the cost" of the environment. The magnitude of inequality at the global level reveals a world that appears severely unjust and excludes many. Confirming the interconnected nature of global inequality, Bourguinon [24] observes that these "other" forms of inequality are highly correlated with differences in per-capita income across countries. He further alarmingly asserts that "this inequality condemns nearly half of humanity to poverty and has made survival itself precarious for more than a fifth of humanity" (p. 24). Therefore, when we answer the fundamental question of global inequality, "The inequality of what?", the answer is a fuller picture than simply income disparities; it is the "inequality of people's capabilities" across countries, also reflected in global inequalities of wellbeing.

### 3. Drivers and Outcomes of Global Inequality and Exclusion

Given that a wide variety of causes of economic inequality exist, as denoted by the rich body of literature on the subject, the emphasis here is on identifying those drivers of inequality that have a strong "excluding" effect on people's formal economic participation, and which occurs across countries. They all lead to income and/or non-income differentials between those who are included and those who are marginalized. Hence, in the context of inclusive economics, the following drivers of global inequality and exclusion are identified (in no particular order, and not exhaustively):

- Legacy of unequal historical patterns: The consequences, and in some cases the continuation, of historical inequalities or injustices of the past within and between countries (e.g., colonization) make this a very underestimated driver of exclusion. Unequal distribution of assets (e.g., land, water or economic capital), wars that led to the unfair redistribution of land ownership, concentration of power resources in the hands of village landlords, and systems of ethnic segregation (e.g., apartheid) all have such marginalizing impacts that it takes generations to rebalance conditions [20].

- "Perverse growth": This is growth that undermines rather than enhances the potentialities of the economy for long-term growth and development. It leads to the exclusion of some people, the concentration of wealth, and very often the maldistribution of resources [32]. It also leads to a consumption-based development model—dependent on increased consumption and standardization—as opposed to a wellbeing-based model, which includes and empowers people to make objective decisions linked to their values and motives. Such a growth-driven economy follows a vertical structure, assuming a trickle-down effect through a separation of production and consumption. Increased inequality is most often a result [33]. Perverse growth is also linked to the unequal ownership of capital, which can be either privately or publicly owned. In terms of global wealth inequality, the 2018 World Inequality Report [34] found that "the combination of large privatizations and increasing income inequality within countries has fueled the rise of wealth inequality among individuals" (p. 16).

- Market fundamentalism: Piketty [35] demonstrated that, without government intervention, the market economy tends to concentrate wealth in the hands of a small minority, causing a rise in inequality. Fukuyama [36] expresses concern that neoliberalism helped generate some of the problems of "failed states" (p. 163) that emerged in the 1990s. The growing problems of governance, lawlessness and the breakdown of administrative competence can hardly be separated from state-neglected uneven development [25]. By reducing government intervention (e.g., tax and regulation to keep inequality in check), market fundamentalism is seen as a contributor to the inequality problem [37]. Ironically, the liberalization of markets has the capacity to obliterate the social capital essential for the long-term sustainability of capitalism itself.

- Globalization: All meanings of globalization suggest some relationship to inequality—at least in terms of heightening it. The United Nations [38] stresses that "the costs and benefits of globalization are not equally shared among countries and peoples" (p. 5). Concerned about how globalization exacerbates inequality, Robinson [39] emphasizes: "Expanding poverty; inequality, marginality; and deprivation are the dark underside of the global capitalist cornucopia so celebrated by the transnational elite" (p. 168). In *The Globalization of Inequality*, Bourguignon [24] more moderately recognizes the significant role globalization plays in the development of inequality, increasing it in most countries over recent decades. The globalization of trade, for instance, has had a substantial impact on the distribution of income, lowering the wages of unskilled labor in developed countries in the face of direct competition from cheap labor costs in emerging economies, and increasing the profits and remuneration of highly skilled labor across the world. It is furthermore undeniable, as the ILO [40] points out, that wealthier countries exploited the benefits of globalization as they possessed a "strong initial economic base, abundance of capital and skill, and technological leadership" (p. 37).

- Imbalanced global economic governance: Perpetuating global inequalities associated with globalization, the International Monetary Fund (IMF), the World Bank and the World Trade Organization (WTO) remain plagued with criticism: the structural underrepresentation of the global South; undermining of democratic ownership; biased and inconsistent decision making; effective impunity regarding harms caused; and a largely unfair international trading system [41]. It is well recognized that the neoliberal structural adjustment policies that were required by the World Bank and the IMF in the 1980s and 1990s from developing countries did much damage to their economies [42]. The ensuing social costs, slowdown in economic growth, and rising inequality eventually put them at a grave disadvantage. In the same vein, the WTO is criticized for enabling exploitation by multinational companies (MNCs) of local industries and firms, thus reducing cultural diversity and economic inclusivity.

- Unequal access to education: The impact of education disparities starts even as early as early childhood education. Since this is largely determined by parents' socioeconomic status and a child's health, only 20% of children in the developing world receive it [27]. Starting from this disadvantaged position, inequality is then often perpetuated by inadequate schooling facilities, insufficient teaching capacity, and many other factors limiting access to education. In many cases, it even increases the risk of parents transmitting poverty to their children by withdrawing them from school and permitting child labor for survival. It is clear that education outcomes severely affect people's productivity and ability to earn a decent income. Since inequality tends to accumulate through life, it affects the human capital development of a nation and eventually its economic ability to compete globally. Being poorly staffed and underfunded, poor quality public services in general drive inequality in many countries [28].

- Social instability: Severe economic disparity is considered to be one of the major global risks in the 21st century, particularly for its link to social problems and instability, such as violent crime, civil unrest and even mental illness [43]. This is especially the case in countries with social structures that overtly foster unequal power relations. Even in the developed world, the increased social dislocation experienced by many, together with growing threats to people's security (e.g., terrorism), adds to the sources of social instability. According to Dietz and O'Neill [44], "inequality produces the conditions for social ills, but it also contributes to environmental problems" (p. 90). They demonstrate in their study that nations with greater income inequality have more health problems and more social breakdowns, confirming that less equal societies have a strong tendency to become dysfunctional, thus worsening inequality. Unequal redistribution of income creates further social tensions between different segments of the population. For instance, it is estimated that in Latin America half of the increase in poverty between 1980 and 2000 was due to redistribution of wealth in favor of the richest [37].

- Segmented labor markets: The impacts of minimum wages, layoffs, discrimination against certain groups, and migration have severely deepened inequality. Women are arguably worst affected when labor regulations are mitigated—such as when paid maternity leave and holiday entitlements are removed, or when unpaid care by the government is held back [37]. Add to this the impact of industrial restructuring (downsizing, outsourcing, dismissals) and job security drastically decreases. Furthermore, in conjunction with globalization, the "iron law" of labor cost reduction inherent in capitalist production has contributed significantly to worldwide depressed wages and worsening work conditions [45]. Unemployment itself—not just the loss of income but also its additional effects on society—is a significant driver of inequality. "Employment precariousness" or the lack of a "decent job" are forms of non-monetary inequality [24]. Discrimination in the labor market, including discrimination against migrants, is a form of inequality that is not fully taken into consideration in standard income inequality measurement.

- Uneven policymaking: Concerns range from inequality being explicitly exacerbated by reducing taxes on the rich (weakening the progressiveness of taxation) and curtailing welfare provision for the poorest to subtler forms. Entrenching unfair advantages and the role of political

and economic elites in the capture of political power and policy are known to be reinforcing inequality [37]. Policymaking, where many are excluded from participation, limits the legitimacy, accountability and very often the efficacy of such policies. Public policy choices that create and sustain inequality are particularly worrisome [46]. Over recent decades, a number of reforms have been undertaken—in the name of economic efficiency—that were meant to improve national economies' competitiveness but have often contributed to a rise in inequality [24].

- Climate change: The 2019 Human Development Report demonstrated how climate change and inequalities in human development are intertwined—where climate change is both a driver and an outcome of inequality [27]. It found that global carbon dioxide emissions are highly concentrated: the bottom 50% of emitters account for 13% of global emissions, while the top 10% account for 45%. The disequalizing impacts of climate change are further evidenced in how unmitigated climate change drives inequalities in human development [47]. The impact is shown in two ways: differential exposure and vulnerability. In the context of the former, differences in the environment add another dimension to inequality of opportunity (e.g., healthy versus unhealthy living and working spaces).

- Technological transformation: Ever since the Industrial Revolution, which caused the "Great Divergence" between industrialized countries (producing and exporting manufacturing goods) and others that depended on primary commodities, technology has played a significant role in global inequality. Today, in the age of artificial intelligence and nanotechnology, another major shift is taking place as the rate of innovation exponentially increases in advanced economies while most others are left behind. As part of a "New Great Divergence" in the 21st century, technological convergence is having a profound impact, reshaping income distribution, inequalities in access to technology, employability, and how technology is replacing workers. This digital divide, which is expected to be entrenched by the Fourth Industrial Revolution, is also reshaping economic power (e.g., the dominance of tech monopolies). It is estimated that, by 2030, 70% of the global economic benefits tied to artificial intelligence will accrue to North America and East Asia [27]. In this way, the technological transformation is both a driver and an outcome of inequality.

It is, particularly, the combined effect of these drivers across countries that is having a substantial influence on escalating global inequality and exclusion. Extreme inequality has corrosive consequences, such as deterrence of economic growth, political corruption, weak social mobility (keeping some families in poverty for generations while others enjoy increasing privilege), and health and social problems, that affect the global community as a whole. As the interdependency of economies and societies grow, so do the spillover effects of inequality.

## 4. Inclusive Economic Criteria

To address many of the "exclusion" effects of global inequality and neoliberal capitalism, an inclusive economic framework is examined. Using the description, outline and basic understanding of IE—an evolving concept—considered in Section 2, key pillars in this framework are explored in this section. For this, the following inclusive economic criteria are identified (not in order of importance).

### 4.1. Inclusive Growth

Ali and Son [48] define inclusive growth as "growth that not only creates new economic opportunities, but also one that ensures equal access to the opportunities created for all segments of society, particularly for the poor" (p. 12). It is pro-poor in that it is focused on improving poor people's incomes in both relative terms (poor people's income improves relative to the non-poor) and absolute terms (when less people end up below the poverty line). Inclusive growth is also broad-based, involving more poor/marginalized people in the growth process through employment. It focuses on greater access to non-income aspects of wellbeing, supported by proactive state policymaking and contributions from other stakeholders [20]. It is aimed at ensuring that the fruits of growth be shared

to specifically eliminate poverty and eradicate income inequality. Inclusive growth is thus anchored in (1) high and sustainable growth to create good employment opportunities and (2) social inclusion to provide equal access to opportunities by all. Lastly, inclusive growth also involves "green growth", being a path of economic growth which ensures that natural resources are used in a sustainable manner so as to enable continued human wellbeing as well as innovative exploration for new sources of growth.

### 4.2. Moving Towards a Circular Economy

In contrast to the linear extractive industrial model, a circular economy is regenerative by design to benefit business (profit-making), society (opportunities to all) and the environment (reducing waste and pollution). It is a sustainable "closed-loop" economic system where all "waste" becomes "food" for another process—either as a regenerative resources for nature (e.g., compost) or a recovered resource or a byproduct for another production process [49]. More specifically, it involves three "secondary production" activities: reuse at the product level (e.g., repair or refurbishment); reuse at the component level (e.g., manufacturing); and reuse at the material level (e.g., recycling) [50]. This requires eco-innovations, defined by Prieto-Sandoval et al. [51] as "the production, application or exploitation of a good, service, production process, organizational structure, or management or business method that is novel to the firm or user and which results, throughout its life cycle, in a reduction of environmental risk, pollution and the negative impacts of resource use (including energy use) compared to relative alternatives" (p. 605).

Apart from the environmental benefits, this extension of the product life cycle opens up opportunities for income generation by more participants in the economic process as it stimulates innovation. Additionally, the circular economy involves the implementation of renting models where manufacturers may rent the same product to several clients, thus increasing revenues per unit and reducing the need to produce more to increase revenues. Plus, significant net material cost savings can be accomplished as well as "upcycling" (continual improvement (upgrades and repairs) of existing goods) [33]. As a result, the circular economy gradually decouples growth from the consumption of finite resources while creating new participation opportunities.

### 4.3. Genuine Economic Progress

This comprises GDP growth that is inclusive, GPI growth, inclusive development, and full ecological responsibility. In practical terms, it means an economy that is growing sustainably, is creating new participative opportunities, and is ecologically replenishing. An improvement in genuine economic progress is indicated by the real increase in economic welfare and/or improvement in the wellbeing of a nation/community as a whole [5]. By taking social and environmental factors into consideration, it presents a more accurate picture of progress than simply income growth. For instance, if the economy is growing but poverty and/or unemployment is also growing and/or net natural resource depletion takes place, there is no real/genuine economic progress. It is also about "internalizing" negative externalities in the economy and optimizing positive externalities [52]. In the case of the former, for instance, companies would pay for the costs of pollution they create (e.g., carbon tax). Put another way, genuine economic progress is the net effect of subtracting negative externalities (e.g., social and environmental costs) from growth in GDP and adding positive externalities not measured by GDP [53]. In the GPI context, this is when it is above zero; zero indicates when the financial costs of poverty and pollution equal the financial gains in the production of goods and services—all other factors being constant.

### 4.4. Building a Collaborative Economy

Also referred to as the "sharing economy", this kind of economic thinking is where the emphasis in the marketplace moves from sole reliance on businesses and large companies to consumers also relying on each other to meet their needs and wants. Owyang's [54] collaborative "honeycomb model", consisting of 12 different sharing economy sectors (some include money, goods, food, services,

transportation and space), gives a helpful picture of what it could entail. Referring to its organic nature, he observes [54]: "Similarly, in nature, honeycombs are resilient structures that enable access, sharing, and growth of resources among a common group" (p. 1). According to Fioramonti [33], it means "improving the quality and effectiveness of human-to-human and human-to-ecosystem interactions, supported by appropriate enabling technologies" (p. 13). Such a wellbeing-based model empowers and includes people to make optimal decisions linked to their motives and values. It forms an integrated network that raises the productive participation of community members as new roles, functions and forms of production and economic utility are created. Goods and services are shared in direct trade, thus replacing the need for a mediating company. Convenience, lower prices and inclusive access become the basis for trading, not ownership per se. As Satell [55] suggests, "rather than assets managed by centralized organizations we have ecosystems managed by platforms. Capabilities are no longer determined by what you own or control but by what you can access" (p. 1). Enhanced by technological innovation, such collaborative networks strengthen interrelationships for higher levels of inclusive production and raise a community's social capital—whether it exists online or geographically.

A caring economy emerges where shared values/ethics/norms organically take shape. This becomes the basis for activities such as resource sharing, inclusive business and environmental responsibility. Community collaboration takes the effect of truly caring for others (integrating all members) and living closely to healthy ecosystems. Not just raising the standard of living but raising the quality of life becomes essential. The African concept of "Ubuntu", which necessitates inclusive economic thinking—even in a digital economy—reflects this well in its values: collective benefitting, reciprocity, humanness, security (safety and means) and collective wellbeing.

*4.5. Inclusive Economic Policies*

The role of the government in creating a truly enabling environment is crucial to inclusivity [2]. Efforts by society/civil groups as well as inclusivity endeavors by the private sector must be reinforced and systemically entrenched by government initiatives. Policies actively geared towards generating employment opportunities, poverty reduction, better income redistribution, social justice and environmental mindfulness are paramount. An effective progressive tax system, affordable housing, pro-poor social protection, social security in health and education, and fiscal and monetary policies that favor long-term growth should be vigorously implemented [56]. To assist with tax reform, one interesting area to explore is "smart redistribution". Since technology is increasingly making the transparency of monetary flows possible, a system in which taxpayers would be able to track the spending of their tax payments would result in much higher commitment. Smart redistribution would take this a step further by empowering taxpayers to become actively involved in the welfare choices of society by deciding for themselves where to allocate their contributions [33]. With the state then becoming a "manager" of funding choices offered to taxpayers, the latter are reconnected to the progress and tangible impact of their payments, thus creating incentives to invest more in collective wellbeing.

From the description in Section 2, and these five interwoven criteria, it becomes clear that inclusive economics is more than another theory or concept; it is about holistically reorganizing society in such a way that economic benefits are shared equitably and economic, social and environmental costs are reduced effectively, especially in how it excludes the poor/marginalized. A new functional reality for the economy as fully integrated into society, rather than society (and the environment) as part of the economy, is essential to such a reorientation.

## 5. Economic Sustainability and the SDGs

Since the inception of the SDGs in 2015, a marked ideological shift—albeit small—in economic frameworks towards sustainability was firmly established [47]. While it takes time for economies to adjust, especially given how entrenched neoliberal consumption and production patterns are, the stark realization of the need for change has certainly set in. However, with global inequality and exclusion

still on the rise, it is vital to make sure of the positive impact of the SDGs in economies and for that sustainability be pragmatic. In a comprehensive study, and as arguably an apt representative precis of key sustainability aspects in the literature, Edwards [12] identified five sustainability principles, which will be employed here (1) as a yardstick in linking them with the SDGs and (2) as a way to identify some core elements in building a sustainable economy:

- Community: Sustainability gives recognition to the fact that there is an interdependence at all levels of community—from the local to the global. Sustainable strategies at the local level involve housing, jobs, transportation, healthcare, education and arts. At the regional level, they deal with the impact of neighboring communities and sharing resources and interregional infrastructure. At the national and international levels, sustainable strategies are promoted or hindered by policies on taxes, energy, healthcare, food, etc. Due to their relatedness to the three Es of sustainability (ecology, economy and equity), their interdependence is systemic, as they contend with difficult and interrelated problems at all three levels [12]. Of particular significance is how this creates a platform for integrating common interests and finding cross-sectoral solutions. It is furthermore beneficial that indicators of a sustainable community (or community of communities) can specifically strengthen the areas where the links between the economy, society and the environment are weak. Essentially, all the SDGs adhere to the principle of "community" to a greater or lesser extent [57]. The lack of progress on, especially, Goal 17 (strengthen the means of implementation and revitalize the Global Partnership for Sustainable Development), is significant here. Continuing a downward trend, during 2018, aid to Africa fell by 4% and total Official Development Assistance fell by 2.7% [27]. In addition, private investment flows globally are unsynchronized with sustainable development, while trade tensions, especially between the US and China, are causing a retreat from multilateral cooperation.

- Commerce: Similar to the "precaution principle" instituted by Germany in the 1970s, this sensitizes businesses to rethink their responsibility to society, requiring them to use foresight as regards the community and environmental costs of their business ventures [12]. A new business ethic is taking shape where the responsibility moves from the customer and government regulators to the corporation. Being accountable not only to shareholders but to all stakeholders (partners, employees, customers, supplies and the larger community), it is setting sustainable businesses apart in a "Conscious Capitalism" movement. Significantly, the move to a willingness by companies to achieve development objectives by integrating them into their core business models, and not just add-on corporate social responsibility, is what is truly setting a new standard [58]. Increasing evidence, also, of inclusive businesses outperforming traditional competitors are incrementally changing "business as usual" for good. Plus, as a driver of innovation, they have started exploring how the power of business can solve social and environmental problems. The SDGs that can be directly linked to this principle are goals 5 (gender equality), 8 (decent work and economic growth), 9 (industry, innovation and infrastructure), 12 (responsible consumption and production), and 17. Of particular significance are goals 8 and 12. The annual growth rate in real GDP in the least-developing countries is continuing a downward trajectory (from over 8% in 2007 to just over 4% in 2017), with the global unemployment rate at 5% in 2018 [54]. The global material footprint—the total amount of raw materials extracted to meet final consumption needs—continues to grow rapidly (92 billion in 2017), outpacing population and economic growth.

- Natural resources: Valuing the integrity of the land and its resources is crucial. While principles in this area reflect the challenges faced by resource extraction industries (e.g., minerals, fish, oil and gas, and land for agriculture) to implement sustainable strategies, the need for them to retool and search for alternative ways to manage their operations has become immutable [59]. Two key variables stand out regarding this principle: (1) assessing whether a resource is renewable or not (if not, it requires recycling of existing materials and moving to renewable alternatives) and (2) short-term versus long-term perspectives (short-term profits often bring resource destruction while long-term sustainable practices require a more comprehensive strategy) [12]. These being

global issues, it is often the local dimension that is most affected. For instance, the positive impact of agribusiness on family farms and communities highlights the value of sustainable farming practices. In terms of the SDGs, goals 7 (affordable and clean energy), 9, 12, 13 (climate action), 14 (life below water), and 15 (life on land) are relevant here. In 2016, 17.5% of total final energy consumption came from renewable energy (up from 16.6% in 2010). Furthermore, the proportion of fish stocks at biologically sustainable levels declined from 90% in 1974 to 67% in 2015 [27]. Between 2000 and 2015, land degradation affected one fifth of the Earth's land and the lives of one billion people, resulting in a considerable loss of services essential to human wellbeing.

- Ecological design: This principle relates to the interdependence between human environments and ecosystems. More specifically, finding alternative building strategies that are in harmony with communities and ecosystems is essential. Few are aware that, globally, buildings use 40% (three billion tons annually) of all raw materials [60]. Only in the US, buildings are responsible for over 30% of total greenhouse emissions, almost 150 million tons per year of demolition and construction waste, and 15% of potable water use [12]. "Green" building strategies have the added benefits of reducing operating costs, improving people's health and safety, and enhancing the quality of life in communities. Conveying an innovative fusion of biology and engineering, the principle of ecological design focuses on the interaction of architecture, people and nature. While more SDGs may be indirectly affected by this principle, it is primarily goals 9 and 11 (sustainable cities and communities) to which it contributes directly—both positive and negative. According to the United Nations [27], "total official flows for economic infrastructure in developing countries reached $59 billion in 2017, an increase of 32.5% in real terms since 2010" (p. 40). Yet, in 2018, due to urbanization, one out of four urban residents around the world lived in slum-like conditions. This is outpacing the construction of adequate and affordable housing.

- The biosphere: The relationship between humans and nature is at the heart of sustainability. This principle concerns itself with the following central question, as pointed out by Edwards [12]: "How can we live in harmony with the natural world and create a healthy and vibrant economy that supports all life on the planet?" The role of nature as model and teacher to guide human actions also underlies this emerging environmental ethic. It calls for heightened sensitivity of our human impact and responsibility since we and all other species depend on the ecosystems in the biosphere for survival. Following "biomimicry" principles, an increasing number of businesses are giving serious consideration to nature as a beneficial model for business and industry [61]. There is general consensus that all the SDGs are focused towards addressing the central question above. The ones that can be directly linked to the biosphere principle are goals 6 (Clean water and sanitation), 7, 13, 14 and 15. It is especially disturbing that two out of five people worldwide do not have a basic handwashing facility with soap and water [27]. Moreover, biodiversity loss is still accelerating as the risk of species extinction has worsened by almost 10% since 2000. This raises the important connection that Costanza and Kubiszewski [47] express: we are "fighting poverty by healing the environment" (p. 293).

It is notable that, although these five core principles of economic sustainability are concerned with a number of the SDGs directly, over a third of them are still somewhat excluded from being directly and assertively addressed. Examples include goals 1 (no poverty), 2 (zero hunger), 3 (good health and wellbeing), 4 (quality education), 10 (reduced inequalities) and 16 (peace, justice and strong institutions). In light of this, the importance of bringing "economic inclusivity" principles into the equation could help to close the circle, as it were.

## 6. Integrated Sustainability Priorities for Better SDG Adoption by Countries

One aspect that stood out clearly from the 2020 global health crisis is the need for collaboration: all stakeholders had to play their part in overcoming the threat and consequences of the coronavirus turmoil. One common denominator between the SDGs mentioned above—particularly the ones listed last—is that they all require the need for intense collaboration. In view of global inequality being

identified in this study as the main source of economic "exclusion" and the reality that the global economy increasingly functions as a collective, the emphasis is shifting more and more towards economic collaboration. Our interdependent reality demands a rethink of how economic participation and resources should be managed—as a collective. The goal remains twofold—like two sides of the same coin—equal opportunities for all and equitable economic outcomes. The central problem is also twofold: inequality in human capabilities and ecological limits [22]. The solution lies in the effective collaboration of key stakeholders in taking us closer to answering this fundamental question: How can we live in harmony with the natural world and create a healthy and vibrant economy that supports all life on the planet? Sustainability priorities are therefore identified in an integrated framework—including aspects of economic sustainability, the SDGs and inclusive economics (IE)—to mitigate the negative impact of global inequality while also improving countries' adoption of the SDGs. It should be clarified at this point that, although economic integration may increase inequality, especially if it is "selective integration" (e.g., between rich nations/stakeholders, at the exclusion of poor nations/stakeholders), the emphasis here is "non-selective" integration.

The key to a collaborative framework is reciprocity. Once all stakeholders come into agreement on the primary objective and all understand their roles, they can work from different sides to achieve their goal. A pragmatic approach is essential. Arguably, the challenge with the limited success of the MDGs and the SDGs is the lack of ownership (buy-in) of them by all stakeholders in society. While they are inclusive in nature, they seldom engender inclusive collaboration. The needed "inclusive commitment" can be brought about by IE. A first priority is establishing the "common interest" that government, civil society and businesses share. Inclusive policies, inclusive community functioning and inclusive business need to meet each other and come into agreement on how each party can contribute to enhance collective wellbeing. From all three sides, the goal of creating equal opportunities so as to improve the wellbeing of all is pursued in a process of reciprocal benefit sharing. This opens the door for grassroots adoption of the SDGs. Micro-level impacts would include investing in a community's social capital and renewable capital by the private sector; "green" building strategies through private–public partnerships; state-funded skills development projects determined by a combination of the needs of the community and its businesses; direct trade opportunities and resource sharing/renting; entrepreneurial ventures by the community for recycling, upgrades or upcycling; and the state creating incentives to invest more in collective wellbeing. As this "public–civil–private" partnership grows, government, communities and businesses can then continually explore new ways to innovatively enable increased participation, involving adaptive learning processes that are organic and flexible to changing conditions, yet that are strongly focused on positive outcomes.

Such an economy of care (from the bottom up) can then take effect from the "top down" on a macro level as well, giving credence to the second priority: a shared national commitment to create an inclusive enabling environment that stimulates genuine economic progress at all levels of the economy. Most significant would be the realization of inclusive growth to reduce disadvantages and create opportunities for poor people's incomes to rise faster than those of the non-poor, while optimizing production and productivity. Another element to fulfilling the second priority is the actualization of inclusive governance. Greater public participation in policymaking and choosing policy preferences, as well as increased transparency through the use of technology (e.g., smart redistribution), would increase the success rate of policies and help meet the SDG targets. Civil society participation would also assist with negating the dominance of MNCs/monopolies and supporting the eradication of labor market discrimination. A third element would be a comprehensive strategy for inclusive education: government providing quality education to all; communities protecting and upgrading school facilities and parents becoming more involved in their children's education; stimulating and facilitating skills development (e.g., for recycling) in communities; educating people on the balance between economic and ecological sustainability; using technology to make learning more effective; etc. A new culture of learning is instrumental to involving more role players—from the

public to private sector—and for holistically reorganizing society with a deeper understanding of the need for and value of an inclusive economy.

Collaboration deepens integration. The third priority involves the need for international cooperation as nations recognize the limits to economic growth and assist each other to develop steady-state economies. It remains an open question whether the world's resources can afford developing countries developing along the same growth paths as developed countries. [24] Nations have arrived at a precipitous point on the edge of breakdown [62]. Unprecedented collaboration is necessary to reconfigure international trade and for developing economies to identify alternative paths to increase their citizens' wellbeing, which will better enable SDG adoption [44]. At the same time, wealthy high-consuming nations must take greater responsibility for the impacts of their consumption and make robust adjustments to their growth patterns. They also need to collaborate with low-developed and developing countries to "make room" for their growth and allow them to co-exist and co-develop; plus, they must demonstrate leadership on technology transfer to help fast-track inclusive development. In the same context, international organizations such as the IMF, World Bank, WTO and United Nations need urgent democratization to ensure fair representation, especially of developing countries with high population numbers. The importance of the increased role and impact of civil society movements, as "governance from below", is also vital in shaping participative global economic governance. A new direction in the global policy agenda towards a mindset of much closer cooperation is necessary to ensure that the world be viewed as heterogeneous societies and cultures whose fates are intertwined, rather than just a collection of individual countries competing against each other for dominance. Importantly, as Dietz and O'Neill [44] point out, "nations need to find the political will to maintain checks and balances on economic scale and power. Now is the time to cultivate national temperance and intensify international cooperation" (p. 190).

To make these three priorities a reality in their full sense and strengthen SDG adoption, one fundamental requirement is needed: a deep commitment among stakeholders at the local, national and global levels. Arguably, the challenge of our time is that the "social contract" that holds society together is broken. Growing concerns about rising inequality, jobless growth, social unrest and poor global governance are ruining the social contract between government, business and citizens. The deeper issue is the belief that leaders have betrayed public trust and that systems are unfair [43]. The Pew Research Center found that trust in governments is at its lowest levels in 50 years [63]. Similarly, the credibility of corporate chief executives is plummeting. People have lost trust in politics and business, while public scrutiny via social media of companies, institutions and governments is growing. Ironically, although people are empowered by this radical transparency to expose unethical behavior, they experience a disconnect. This lack of trust diminishes people's sense of participation in society as well as their social responsibility [43]. It isolates people, resulting in them caring more about survival than solidarity.

To overcome the broken social contract and meet the fundamental requirement mentioned above, a new "social covenant" is needed. Whereas the old "contract" model is primarily based on "rights", a social covenant formulates a deeper connection since it is also based on values and trust. A contract is merely transactional, but a covenant is morally binding. In this regard, the World Economic Forum [43] identified three universal values: (1) "the dignity of the human person, whatever their race, gender, background or beliefs; (2) the importance of a common good that transcends individual interests; and (3) the need for stewardship—a concern not just for ourselves but for posterity" (p. 1). While social covenants can differ between countries and groups, this answers people's demands everywhere for new, more transparent, collaborative and inclusive values/norms that prioritize wellbeing, happiness and meaning as much as profits [43]. Revealing the changing nature of the world, the idea of a "just economy" and the meaning of a "moral economy" are openly discussed in the public arena. A new social covenant between citizens, businesses and governments is part of a "great transformation" taking place, which reflects a general consensus to move away from a narrowly defined shareholder economy towards a stakeholder economy, where the interests of everyone—even the environment

and that of future generations—are included (IE). Shared values/norms/ethics would then guide decision-making at all levels as room is made for moral courage in the economy. Like ecosystems, economies need a balance between efficiency and resilience. Social covenants will raise the efficacy of global and local partnerships, resulting also in greater transparency, accountability and democracy in global governance [64]. This truly charts a new path away from what Piketty [35] calls "patrimonial capitalism", towards "inclusive capitalism".

## 7. Conclusions

A rethink of today's economy must start with a re-appreciation of the original meaning of the word "economics": the management of the household's resources. Derived from the 15th century Greek word, *oikonomos*, the emphasis is on a fair distribution of resources, optimal productivity and restrained accumulation of wealth [65]. Six centuries later, the "global family" is in need of realigning the global economy with its original intention. The very word "economic" might say it all. The cumulative co-existence of excessive luxury and starvation in a Pareto-optimal global economy overreaching its ecological limits has led to wide concern but little change. Klein [66] calls it "a civilizational wake-up call" which is "spoken in the language of fires, floods, droughts, and extinctions—telling us that we need an entirely new economic model and a new way of sharing this planet" (p. 235). The World Economic Forum [66] confirms that "something has gone terribly wrong in our societies" (p. 2). During each recurring decade since the end of the two world wars, the global community has witnessed growing inequality [21]. Questions such as "what is fair, just and right" are taking economics sensibly in a moral direction—especially as many cases of economic inequality are fueled by corruption and greed. Three reinforcing factors that have arguably contributed most to this dangerous development path are (1) the monist ideological dominance of the neoliberal capitalist paradigm; (2) exploitative profit-driven business practices; and (3) neoliberal public policies that have failed to properly attend to the poor and that have created more scope for the concentration of wealth.

The fact that inequality, even more than poverty, harms everyone in society, points to the need for new perspectives on economic growth and progress [67]. The case for pluralism in economics is strengthened by the current neoclassical hegemony's "inability to take account of issues of persistent poverty, inequality and environmental degradation and destruction" (p. 1126). McGregor and Pouw [3] add, further, that "when we focus economics on human wellbeing, then the moral case for pluralism in its own right becomes clearer" (p. 1127). Inclusive economics does exactly that. Most probably, the primary challenge with why the SDG progress is slow and why global inequality in all its dimensions persists is because there is no sufficient "context" created for change to truly take root. The environmental crisis and the social crisis are still "hidden", largely unnoticed by the forces that drive growth, profits and supercapitalism. The reconstruction drive after the Second World War motivated worldwide cooperation because the crisis was clear. Therefore, to create sufficient context for change, firstly, humanity's current existential crisis needs to be brought into the foreground more clearly. Perhaps, the 2020 global coronavirus crisis and its economic consequences will contribute to fully revealing our true state, since "life as we know it" is arguably over.

Secondly, people, corporations and governments need to be fully educated (fourth E) on the reality and implications of living in an interdependent world. Clarity is needed as to why this integrative reality must be met with inclusive frameworks for the economy, society and its governance. This study has attempted to bring more clarity in all of these areas to, ultimately, take us further on the road towards genuine economic progress (this also includes higher SDG adoption rates by countries and drastically reducing global inequality (exclusion)). Therefore, in redirecting towards inclusive economic sustainability, the study identifies the following principal determinants for change:

- Full integration into business: a shift from corporate social responsibility to inclusive business;
- Enabling society to take initiative: a shift from reactive to proactive community development;
- Steering a new development path: a shift to inclusive growth through interactive governance.

Arguably, the key to sustain the impact of these factors of change is a new social covenant or covenants, which are necessary to strengthen commitments by stakeholders in the move to a stakeholder economy. Deeper commitments based on shared values/norms/ethics between citizens, governments and corporations—especially at community level—are fundamental to ensure the genuine evolution of the economy out of "predatory capitalism" and into "inclusive capitalism". In this way, sustainability becomes a main driver of innovation, ensuring that social equity and ecological parity not only raise the standard of living, but raise the quality of life. While further research is needed regarding the relationship between sustainability and economic inclusivity, this study contributed towards identifying inclusive business, participative community and inclusive public policies as three main requirements—supported by social covenants—for genuine progress.

**Funding:** No funding involved.

**Conflicts of Interest:** The author declares no conflict of interest.

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
