# Peer review of "Inclusive Economic Sustainability: SDGs and Global Inequality"

_sustainability, doi:10.3390/su12135427_

Round 1

Reviewer 1 Report

This paper is essentially an essay in political economy. Personally I do not agree with some ideas expressed and some authors who are cited consider that they make erroneous analyzes. Inequality is not in itself bad ... or what is better, to be equally poor, or unequally rich? Terms like "capitalist neoliberalism" are cumbersome and ideologically charged, when you could just say "liberalism" (it's the same thing but without ideological burden). Furthermore, these terms do not fit with a world year after year with a greater presence of the state and the public budget in the supposedly capitalist economies (United States, Europe, South America, etc.), putting them on the brink of bankruptcy.

Despite these personal insights, I consider it to be an interesting reading work to help reflect and improve the own economic reasoning, whether you agree or disagree with the above, and I recommend making some small improvements:

  • Many references are made to COVID-19 and quite a few are not necessary for two reasons: the topic covered in the article is equally interesting before and after COVID-19; the "revolutionary" impact attributed to it does not fit in with previous pandemics in which more people died (1918-1920 flu; 1957-1959 flu, 1968 flu) or recent epidemics (SARS 2004, 2009 flu, etc.). The only thing new in this case is the radicalism of the actions taken by the administrations.
  • It would be nice if there were any other references in the opening paragraphs of section 2.

Author Response

Dear Reviewer

I wish to thank you for your valued inputs regarding my article. I have tried my best to make the necessary adjustments to accommodate your observations. Please see the attachment.

Reviewer 2 Report

Dear Author,

I read your paper and felt that everything what was done in the world is guilty and everything do inequality and even your article will make us unequal as I have not article in this journal. 

I think you must to improve your introduction by providing more arguments for actuality of the topic. Also you must to provide with information which gap you are going to close and critically assess what have done before you. 

In part 3 of your article Drivers for Global Inequality and Exclusion you must to classify as some of the statements are reasons and other outcomes. Like,  climate change it is outcome and not the driver of inequality. Also try to rethink your judgements like technological transfer. The providing the modern guns for some less development countries do not create equality. it is creates more fear and anxiety in the world like, global terrorism. 

In part 4 Inclusive economic criteria can you think on some more innovative criteria's as now these one's are known. 

Also I do not understood the further fade of Edwards sustainability principles and why you chosen them. What were the arguments and what you consider also for your choice. Pls explain. 

I also do not understood you position from one hand you are against integration and provided arguments in part 3, but in part 6 you are for integration. But integration is very important for globalization and etc.

All the best

Author Response

(The authors gave the same response as above.)

Reviewer 3 Report

The study represents the theoretical approach to research problems that included looking for significant factors of inclusive economic sustainably. The basis for the article include mostly a review of the research result fo other Author; thus, it has a literature review approach. This kind of paper is very important, to sum up, the theory issues with the new trend in the economy. The Author also used macroeconomic data to supports indicated factors in each sector of analyses.

However, the article touches too many different issues that the latter are not related to each other (not in every case). In my opinion, nit would be very beneficial to add some diagrams that will explain and summarised discussed factors in the area of economic inequality, economic suitability and global inequality.

I understand the covid related justification of the study, but later it is not strongly related in presented economic theories. However, I assessed the literature view as very detailed and substantiating the research topic undertaken by Author.

The article strongly suites to journal scope of topics.

Other comments:

Abstract

Please add the study contribution to the literature/theory in the abstract.

Introduction

The theory issues and of the inclusive economy should be added in the introduction part.

It has roots in institutional economy theories, even from 1957.

Line 28 – "brutal health crisis" – maybe it could be named more in "scientific" way

Line 32-33 – here, the SDG from the title is underlined, but after reading the paper, I lost orientation with SDG relations to every discussed aspect.

Line 36 – economic inequality is related to MDG and SDG development – add one sentence how it was updated in the new concept

It could that the Authors joined the inclusive economy issues with development. Please add more up to date references to this concept.

Line 43 – "specific effectiveness" – how to understand that?

Conceptual framework

This part is very important; however, it is also very complicated to read. I recommend making subpart of the text and add diagrams.

Line 81 – "old sector" – what do you understand behind "new sectors"?

Line 93 – add primary research on the GPI indicator.

Line 98-104 – could you add some research conclusion fo Autors on GPI measurement according to emerging and developed economies? This part is interesting, but if I shortened the text, I would cut this part of the text to make it more concise.

Line 114-122 – a very nice summary of GDP measures and GPI.

Line 123 – the second key concept is distinguished like the third, but the first it's not so strongly underline.

Line 139 – three pillars of 3E. You referred to them in conclusion; maybe you could also include them in the diagrams as a background for issued factors.

Line 123-142 – the sustainability definition is correctly described – well done. It useful that later the Author relates it to development that is also related do SDG.

Line 158 – "this lead" …but what? Is it the beginning of a new paragraph? All before mentions parts?

Line 188 – the third key context is underline – please remember about the same style for the first one.

The global inequality is related later to differences across nations (line 202-219) – then to the norms (line 220-228)  and income issues (line 229-246). It would be easier to understand if the structure of the described dependencies had been previously drawn up.

Drivers of Global Inequality and Exclusion

I highly assessed this part of the text.

I propose to add diagrams visualizing the described global inequality areas.

Inclusive Economic Criteria

I highly assessed this part of the text.

I propose to add diagrams visualizing the described global inequality areas; it will generate the added value of the text.

Integrated Sustainability Priorities for Better SDG–Adoption by Countries

I highly assessed this part of the text.

Conclusions

Please emphasize contribution to theory more strongly.

It would be good to add a methodology 3-4 sentence for the study. Please also add future research in that filed and limitation fo the study.

Author Response

(The authors gave the same response as above.)

Round 2

Reviewer 2 Report

Hi,

Ok, thanks!